# Label-free imaging of immune cell dynamics in the living retina using adaptive optics

Aby Joseph[1][†][*], Colin J Chu[2][†], Guanping Feng[3], Kosha Dholakia[4], Jesse Schallek[4,5,6]

[1]The Institute of Optics, University of Rochester, Rochester, United States; [2]Translational Health Sciences, University of Bristol, Bristol, United Kingdom; [3]Department of Biomedical Engineering, University of Rochester, Rochester, United States; [4]Flaum Eye Institute, University of Rochester, Rochester, United States; [5]Department of Neuroscience and the Del Monte Institute for Neuroscience, University of Rochester, Rochester, United States; [6]Center for Visual Science, University of Rochester, Rochester, United States

**Abstract** Our recent work characterized the movement of single blood cells within the retinal vasculature (Joseph et al. 2019) using adaptive optics ophthalmoscopy. Here, we apply this technique to the context of acute inflammation and discover both infiltrating and tissue-resident immune cells to be visible without any labeling in the living mouse retina using near-infrared light alone. Intravital imaging of immune cells can be negatively impacted by surgical manipulation, exogenous dyes, transgenic manipulation and phototoxicity. These confounds are now overcome, using phase contrast and time-lapse videography to reveal the dynamic behavior of myeloid cells as they interact, extravasate and survey the mouse retina. Cellular motility and differential vascular responses were measured noninvasively and in vivo across hours to months at the same retinal location, from initiation to the resolution of inflammation. As comparable systems are already available for clinical research, this approach could be readily translated to human application.

**\*For correspondence:**
aby.joseph@rochester.edu

[†]These authors contributed equally to this work

## Introduction

A robust and noninvasive method for directly imaging immune cells in humans has not been established, limiting our understanding of autoimmune and inflammatory disease. Immune cells typically provide weak optical contrast requiring exogenous or transgenic labeling for detection even in animal models. The eye is optimally suited to image immune responses in vivo (*Rosenbaum et al., 2002*). As the only transparent organ in mammals, no inflammatory surgical or environmental perturbation is required. However, aberrations of the otherwise clear optics of the eye limit achievable image resolution. We employed a custom adaptive-optics-scanning-light-ophthalmoscope (AOSLO) to image the mouse eye. By correcting for the eye's aberrations, single-cell resolution provides detailed imaging of photoreceptors to erythrocytes (*Joseph et al., 2019*; *Liang et al., 1997*; *Roorda and Duncan, 2015*). Incorporating our recent advance (*Guevara-Torres et al., 2020*) on AOSLO phase contrast approaches (*Chui et al., 2012*; *Scoles et al., 2014*; *Sulai et al., 2014*; *Guevara-Torres et al., 2015*; *Guevara-Torres et al., 2016*) allowed deep tissue detection of translucent immune cells. We combined and applied these strategies and serendipitously discovered that immune cells and their dynamics could be imaged without labeling using 796 nm near-infrared light, to which the eye is less sensitive, and at far lower levels (200–500 µW) than multiphoton systems, which can be phototoxic (*Galli et al., 2014*). This new approach builds on our ability to quantify

single-cell blood flow in vessels (*Joseph et al., 2019*) revealing the dynamic interplay of blood flow and single immune cells in response to inflammation in the living eye. Our approach is in a similar power range to those already employed safely in human AOSLO studies, where phase contrast approaches have also been used to study other retinal cell types (*Scoles et al., 2014*; *Rossi et al., 2017*). This speaks to the feasibility of translating this technique to the clinic.

## Results

To model an immune response, ocular injection of lipopolysaccharide (LPS) was used to provide an acute but self-resolving inflammatory stimulus: endotoxin induced uveitis (EIU) (*Chu et al., 2016*). Potential immune cells were observed adjacent to retinal veins only after image registration, frame averaging and time-lapse imaging (*Figure 1A*, *Video 1*). Membrane remodeling, pseudopodia formation and motility (consistent with immune cell structure and function) were visible, distinct from static neurons or macroglia (*Figure 1B* and *Video 2*). Within post-capillary venules, leukocyte rolling, crawling, and trans-endothelial migration behaviors were detectable (*Figure 1C*, *Videos 3* and *4*). Heterogeneity in cell distribution, size and morphology was imaged with multiple cell types in different stages of interaction (*Figure 1D*).

We verified these cells comprised neutrophil and monocyte populations by fluorescent marker co-localization. Simultaneous phase contrast and confocal fluorescence AOSLO revealed most leukocytes rolling along venular endothelium were neutrophils using intravenous anti-Ly6G fluorescent antibody labeling, although the proportion may be higher as only 10% of circulating leukocytes may be labeled using this method (*Bucher et al., 2015*; *Figure 1E*, *Video 5*). Conversely, $CD68^{GFP/+}$ mice distinguished a population of cells were infiltrating monocytes and macrophages present both in vessels and extravasated into retinal tissue (*Figure 1F*). More cells were visible using phase contrast than by fluorescence labeling, demonstrating its utility for comprehensively detecting diverse and mixed cellular populations. Tissue resident myeloid cells were also visible by AOSLO phase contrast even in healthy eyes without LPS injection. These were confirmed as microglia or hyalocytes by colocalization of $Cx3cr1^{GFP/+}$ and $CD68^{GFP/+}$ fluorescence (*Figure 1G–I*; *Lazarus, 1994*). Phase contrast even revealed subcellular features, including structures that could represent internal processes such as endosomes (*Figure 1H*, *Video 6*; *Uderhardt et al., 2019*).

As our approach is uniquely non-invasive, repeated imaging at the same tissue location permits longitudinal study throughout the initiation, peak and resolution of an immune response across hours to months within individual eyes (*Figure 1J*, *Video 7*). To quantify immune cell behaviour in these studies, we had to distinguish immune cells from surrounding tissue by using semi-automated deep learning strategy (*Falk et al., 2019*). This correlated well with counts made by masked human observers ($R^2$ = 0.99, p=0.004, *Video 8* (final segment)).

Immune cell metrics were quantified in six mice over five timepoints following LPS injection (*Figure 2A–C*). Compared to baseline ($28.9 \pm 34.1$ cells/mm$^2$, Mean $\pm$ SD) a seven-fold influx of cells was detected by 6 hr post-injection ($208.3 \pm 108.6$ cells/mm$^2$) rising to over an 18-fold increase by 24 hr ($510.4 \pm 441$ cells/mm$^2$) before returning toward baseline at 72 hr ($59.0 \pm 27.8$ cells/mm$^2$) and 10 days ($69.4 \pm 41.7$ cells/mm$^2$). AOSLO also allowed cell motility quantification with maximum cell displacement observed at 6 hr ($16.1 \pm 9.9$ μm, n = 12 cells). Despite peak infiltration at 24 hr, motility was greatly reduced ($4.6 \pm 4.3$ μm, n = 58 cells), best appreciable by longitudinal imaging, consistent with Resolvin-mediated suppression of chemotaxis (*Schwab et al., 2007*).

As retinal tissue is not depressurized by this intravital system, true vascular alterations arising from inflammation can be isolated and correlated to simultaneous immune cell measurements. Adapting our recent work (*Joseph et al., 2019*), red blood cell (RBC) velocimetry, vessel dilation and flow-rate changes were quantified in this same cohort of mice (*Figure 2D–H*). AOSLO revealed micron-level vascular dilations and heterogeneous changes in blood flow in arterioles and venules in response to LPS. Total blood flow increased in the retinal circulation, yet elevated flow in arterioles and venules was achieved in fundamentally different ways. Venules dilated on average 36% (±8%) at 24 hr post-LPS injection facilitating a total flow increase of 67% (±27%) relative to baseline. Conversely, arterioles also showed a total flow increase, however with minimal dilation and a dramatic elevation in RBC velocity (48% (±31%) increase in arteriole RBC velocity at 24 hr) (*Figure 2D–F*).

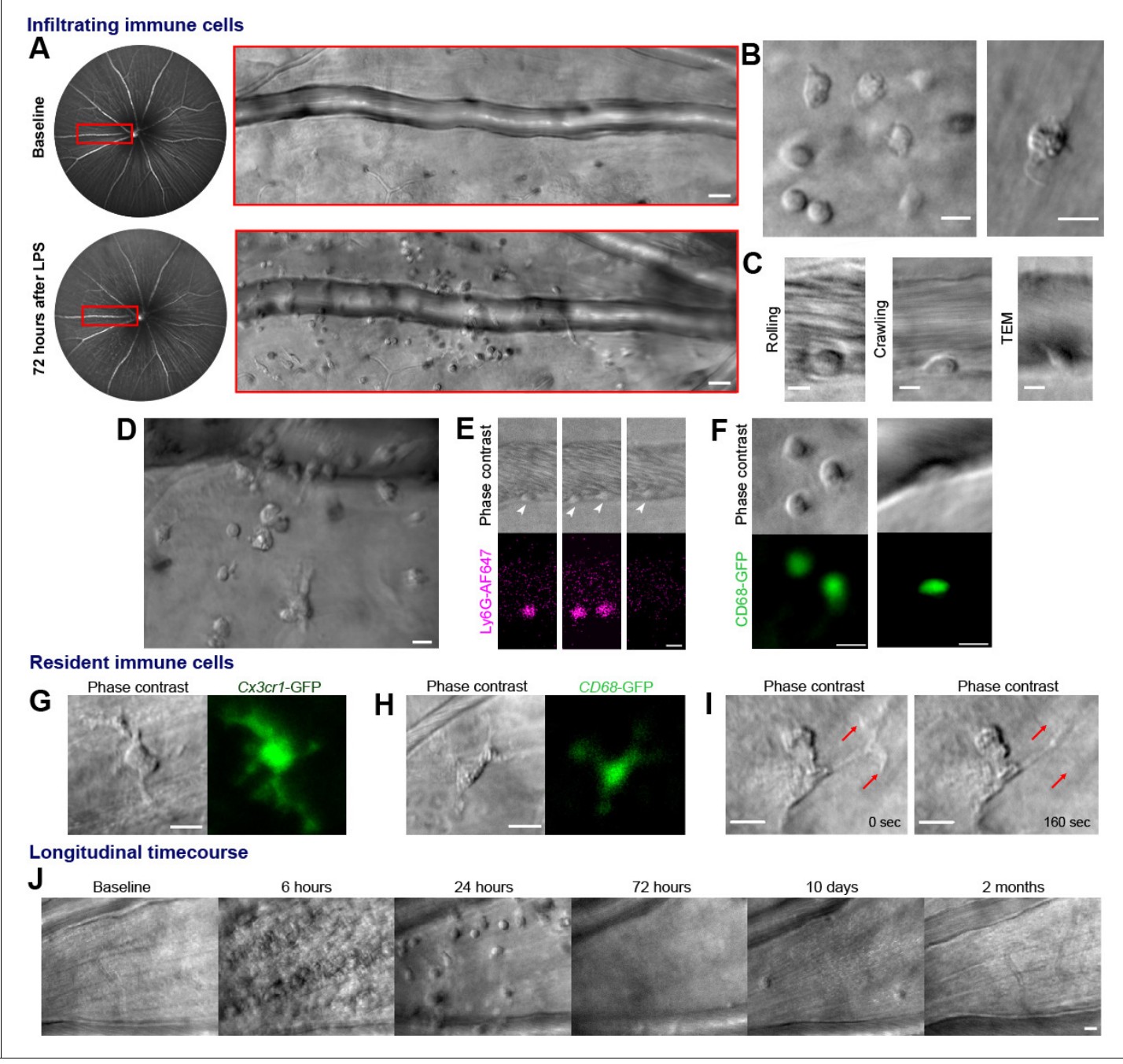

**Figure 1.** Label-free adaptive optics imaging of infiltrating and tissue resident immune cells in the retina. (A) Widefield image showing venule baseline and 72 hr after LPS injection. AOSLO montage (*red rectangle*) detects dispersed immune cells. (B) Detail of heterogeneous immune cells. (C) Intravascular trans-endothelial migration (TEM) stages are visible. (D) Field adjacent to vein 72 hr after LPS. (E) Simultaneous phase-contrast and anti-Ly6G fluorescence reveals leukocyte rolling (*arrowheads*) and (F) *CD68*-GFP reporter shows extravascular and intravascular cells. (G) Representative examples of tissue-resident myeloid cells from *Cx3cr1*-GFP and (H), *CD68*-GFP reporter mice showing colocalization of fluorescence with label-free cells. (I) Phase-contrast image of process remodeling. (J) Longitudinal imaging (hours-to-months) at same location following LPS injection. Scale bars = 10 µm, except in A = 50 µm. Representative images from cohort of eleven (A to D, J) and three C57BL/6J mice (E), three CD68-GFP mice (F and H) and four Cx3cr1-GFP mice (G and I).

The online version of this article includes the following figure supplement(s) for figure 1:

**Figure supplement 1.** Time lapse imaging with label-free AOSLO phase-contrast (796 nm) shows single immune cells migrating (yellow), extravasating (red) and re-entering (green) a retinal venule.

Despite these different mechanisms, conservation of flow was confirmed as arterioles and venules showed correlated changes in flow over time ($R^2$ = 0.83, for linear-fit, *Figure 2G–H*). Velocity, dilation and flow began resolution toward baseline levels by 10 days post-injection.

Thus, we found that arterioles and venules have functionally opposing behaviors when it comes to modulating volumetric blood flow in inflammation, via opposing changes in lumen diameter and blood velocity. Venules modulate blood flow primarily by changing their diameter while arterioles respond primarily not by a change in diameter but by showing increased blood velocity (*Figure 2H*). In our current study, by measuring both blood velocity and lumen diameter, which gives us total flow rate (*Figure 2F*), we build on previous reports that found venular widening in association with inflammation (using diameter measurements alone, and using fundus photography) (*Sun et al., 2009*). However, diameter measurements alone would not have given a complete picture of volumetric blood flow rate and would have missed the functionally opposing behaviours of arterioles and venules shown in *Figure 2H*. In summary, our single-cell velocimetry combined with AO-resolution diameter measurements adds to our understanding of hemodynamic control in response to inflammation, achieved by applying our recently developed blood flow measurement tool (*Joseph et al., 2019*).

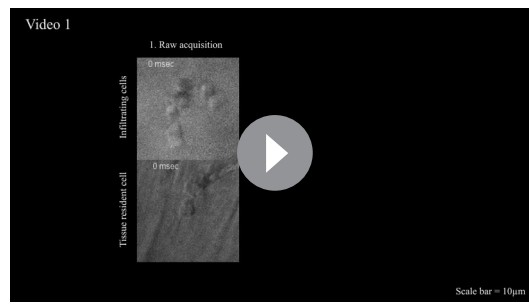

**Video 1.** Demonstration of image post-processing. (a) Raw adaptive optics scanning light ophthalmoscope (AOSLO) corrected 796 nm phase contrast imaging of C57BL/6J mouse retina. Real-time video obtained at 25 fps acquisition (labelled 'raw acquisition') demonstrating movement from respiration and cardiac motion. Image following custom frame-registration. Application of 25 frame temporal averaging and accelerated time-lapse (labelled 'frame averaging'). Top row, cluster of infiltrated immune cells 6 hr post-LPS. Bottom row, tissue resident cell in healthy retina adjacent to a retinal capillary. Scale bars = 10 µm.
https://elifesciences.org/articles/60547#video1

## Discussion

This study advances our understanding of the immune response by imaging both the nuanced differences of vascular perfusion with the first detailed imaging of single immune cell activity imaged without labels. Unlike other approaches, inflammatory surgery is not required; the label-free feature avoids confounds from exogenous fluorescent dyes, gene haplo-insufficiency from transgenic labels and using near-infrared light avoids significant phototoxicity inherent to multi-photon platforms (*Galli et al., 2014*; *You et al., 2018*). This therefore provides the closest fidelity conditions to an unperturbed biological system achievable, which is critical particularly in the context of immune cell behaviours. These advantages are greatest in relation to human translation, where exogenous manipulation and tissue biopsy are often prohibitive. Correlative study using mice to understand AOSLO phase contrast cell morphologies would be highly informative, but there are labeling limitations even in murine systems, and it is rarely possible to label more than three markers simultaneously in vivo. This means populations not already known can be

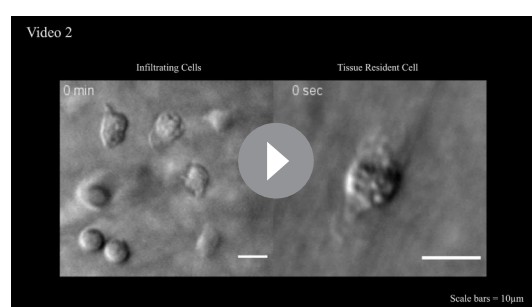

**Video 2.** Label-free AOSLO time-lapse video demonstrating heterogenous immune cell populations and motility. Two magnified locations from *Figure 1b* showing 796 nm phase contrast video acquired at 25 frames/second from two C57BL/6J mouse retinas 24 hr post-LPS injection. The second segment of the video is a full 4.98-degree AOSLO field at a retinal vein 48 hr after LPS injection, revealing a diversity of cell morphologies and motility patterns. Videos have undergone post-processing as described, 25–50 frame temporal averaging. Scale bars = 10 µm.
https://elifesciences.org/articles/60547#video2

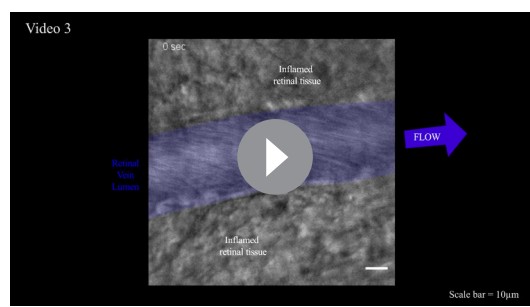

**Video 3.** Leukocyte rolling and crawling in an inflamed retinal post-capillary venule. 796 nm phase contrast AOSLO images taken at 6 hr post-LPS injection with 25 frame temporal averaging. Descriptive overlays provided. Scale bar = 10 µm.

https://elifesciences.org/articles/60547#video3

missed, whereas by phase contrast AOSLO any cell causing reasonable scatter of light could be detected, ensuring a more comprehensive characterization of true tissue infiltration.

Our choice of imaging wavelength (796 nm) provided minimal phototoxicity, greater theoretical penetration through the eye's anterior optics and minimal absorption in the retina without scatter typical of short visible wavelengths. Furthermore, we have previously published a detailed wave-optics simulation to determine the source of contrast in our imaging modality (*Guevara-Torres et al., 2020*), where we found that the combination of this wavelength and other imaging parameters rendered imaging contrast to translucent cells.

Perivascular imaging of immune cells with AOSLO appeared to capture transendothelial migration (TEM). The axial resolution of our system (<16 µm) and known diameters of mouse retinal veins (30–50 µm) (*Joseph et al., 2019*) demonstrates vessel bisection in the majority of images, rather than cells residing above or below the focal plane of imaging. Furthermore, parallel orientation and motility of cells as they interact with the vascular wall would indicate that they are meeting mechanical resistance as they pass through the vascular wall. *Figure 1—figure supplement 1*, second-half of *Video 2* and *Video 4* show that dynamic and motile cells often pause and slow as they meet the vascular barrier; a finding consistent with diapedesis as seen in transendothelial migration. Such behaviors would not be expected if cells were moving above or below the vessel, nor would they be optically visible due to the imaging constraints of the system axial resolution described above.

Consistent with all AOSLO approaches, in our approach, relatively clear optical media is required and severe cataract or vitritis during peak inflammation can limit successful imaging. In our study, this affected 25% of the eyes given EIU. A further limitation is that deeper retinal structures are not as well visualized due to the nature of contrast (*Guevara-Torres et al., 2020*). Moreover, the choroid, which is often involved in ocular inflammation, is challenging to image due to the scatter

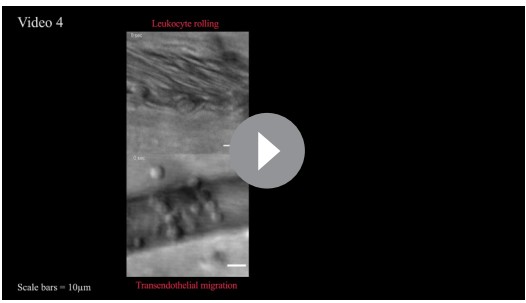

**Video 4.** Examples of diverse immune cell behaviour observable by AOSLO phase contrast imaging. 796 nm reflectance AOSLO images at 6 or 24 hr post-LPS using between 5 to 50 frame averaging. Examples include post-capillary venule leukocyte rolling, transendothelial migration and perivascular leukocyte accumulation, venous leukocyte rolling and crawling with and against blood flow direction, mid tissue infiltrating leukocyte swarming, perivascular cell process contact with intravascular cell and cell migration toward lumen of retinal vein. Scale bars = 10 µm.

https://elifesciences.org/articles/60547#video4

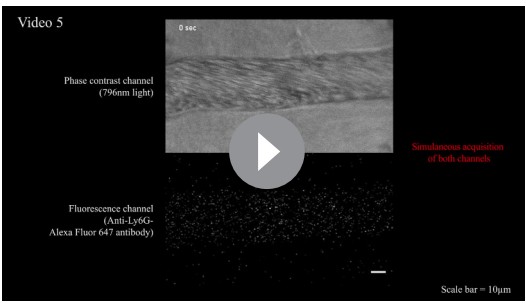

**Video 5.** Neutrophil endothelial rolling within a post-capillary venule confirmed using fluorescent labeling with anti-Ly6G antibody. Representative example of retinal vein imaged six hours post LPS injection. Simultaneous aligned acquisition of 796 nm phase contrast (top panel) and anti-Ly6G conjugated AlexaFluor 647 (positively labeling neutrophils in bottom panel) using confocal AOSLO fluorescence. Scale bar = 10 µm.

https://elifesciences.org/articles/60547#video5

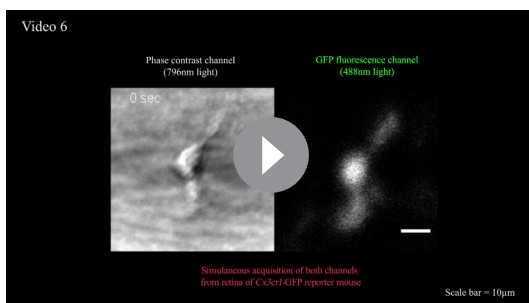

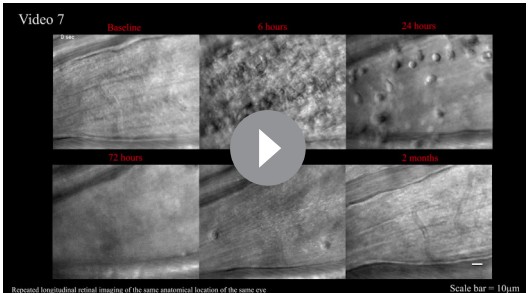

**Video 6.** Cx3cr1-GFP+ tissue resident cell with motile processes. Simultaneous aligned acquisition of 796 nm phase contrast (left panel) and GFP fluorescence (right panel) in healthy retina from Cx3cr1GFP/+ reporter mouse. Second section demonstrates process retraction in tissue resident myeloid cell. Third section illustrates diversity of morphology observed. Forth section highlights visible subcellular features in this population. Scale bar = 10 μm.

https://elifesciences.org/articles/60547#video6

**Video 7.** Repeated longitudinal imaging of the same retinal location from initial inflammation to resolution. Representative recordings from one C57BL/6J mouse eye at the same anatomical location in the retina, identifiable by peripapillary location and capillary and vascular landmarks. Recorded prior to and 6, 24, 72 hr, 10 days and 2 months following a single LPS injection. Scale bar = 10 μm.

https://elifesciences.org/articles/60547#video7

and absorption of the RPE. Nevertheless, our approach here on inner retinal pathology likely extends deep within the retina as work by *Scoles et al., 2017* with a similar detection strategy, has identified a putative macrophage by morphology and motility, deep in the retina of a retinal dystrophy patient. This suggests deeper retinal imaging of immune cells could be possible with further study and refinement. Our work reinforces this exciting finding as we now demonstrate, with transgenic markers in the mouse, that tissue immune cells are reliably detectable by label-free phase-contrast AOSLO.

There are numerous benefits if this approach can be applied to the human eye. Retinal diseases could be characterized for contributions by immune cells for the first time at a quantitative single-cell basis, which has the potential to redefine their clinical status and classification. Active retinal inflammation can only be assessed currently using surrogate biomarkers such as macular edema, vascular involvement or subjective detection of overspill of infiltrating cells into the vitreous cavity. By providing a direct method to diagnose the presence of active immune cells, more accurate diagnosis and differentiation from clinically challenging masquerade entities such as ocular lymphoma would be of great benefit. Finally, accurate monitoring of cellular infiltration to detect early response to treatment could avoid the use of excessive or ineffective therapy in patients with uveitis, yet alone support the screening of novel agents in clinical trials. Furthermore, this could allow study of the many systemic diseases and infections that manifest in the eye. By providing this proof of concept, as AOSLO systems are already available for clinical research, the approach stands to be rapidly tested for translation to human application. Our approach is in a similar power range to those already employed safely in human AOSLO studies, where phase contrast approaches have also been used to study photoreceptors and other retinal cells using a similar light source and wavelength (*Scoles et al., 2014*; *Rossi et al., 2017*). This speaks to the feasibility of translating this

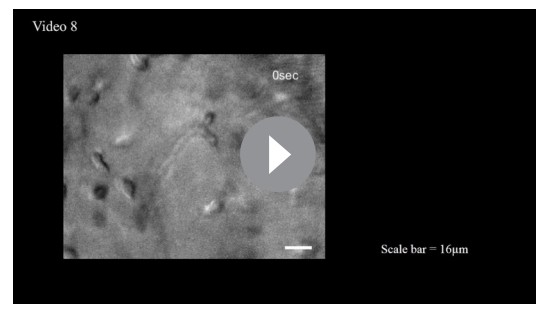

**Video 8.** U-Net cell tracking trace example. Recording from C57BL/6J mouse retina at 6 hr post-LPS injection. Whole field imaged and quantified is shown on the left side. Magnification of one representative cell on the right. Scale bar = 16 μm. Trace is marked with an overlaid magenta line. Second section shows validation of U-Net cell count results against masked human observers across time demonstrating significant correlation.

https://elifesciences.org/articles/60547#video8

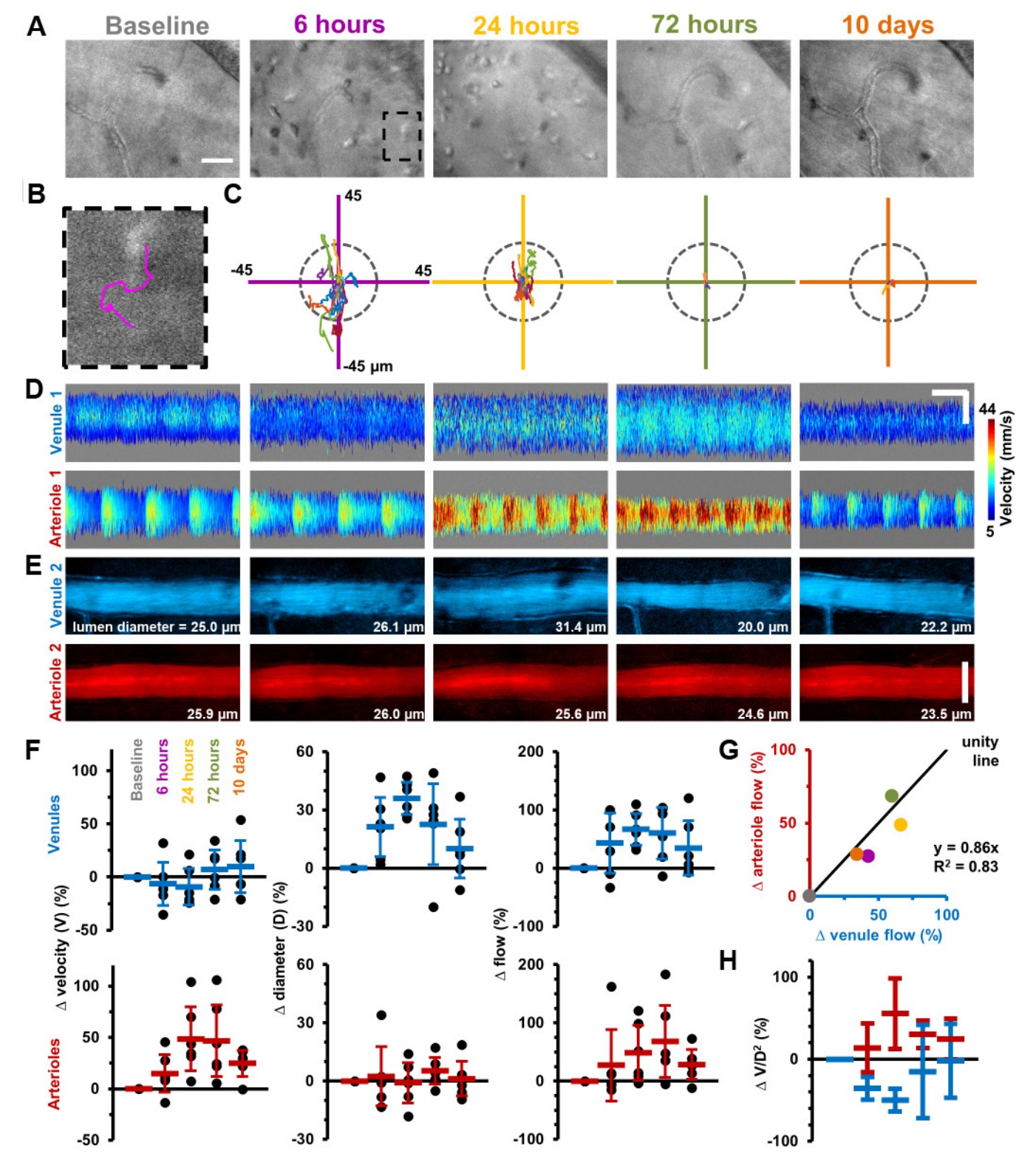

**Figure 2.** Longitudinal non-invasive measurement of combined immune cell dynamics and vascular flow. Measurements of neighbouring venules, arterioles and connecting parenchyma in six C57BL/6J mice given EIU. (**A**) AOSLO phase-contrast images of same region across five timepoints relative to LPS injection, in a representative mouse. Scale bar = 30 μm. (**B**) Magnified image of one cell (marked overlay) with tracked trace (100 s). (**C**) Cell displacements for total cohort at indicated timepoints, in six mice. Displacement traces normalized to each cell starting position. Grey dashes indicate

*Figure 2 continued on next page*

*Figure 2 continued*

radius of typical cell size (13 µm). (**D**) Space-time images with overlaid single-cell blood velocity, in a representative mouse. Arteriolar velocity increases then resolves. Scale bars = 200 ms horizontal, 30 µm vertical. (**E**) Vessel diameter visualized in motion-contrast images of venules (blue) and arterioles (red), in a representative mouse. Venule diameter dilates then resolves. Scale bar = 30 µm. (**F**) Population values of RBC velocity change relative to baseline, lumen diameter and flow rate for venules and arterioles, in six mice. Vein diameter (p=0.008) and artery velocity (p=0.036) exhibit significant changes across time (Friedman test, n = 6 mice). Mean+ SD shown. (**G**) Correlation of change in flow between arterioles and venules (colors correspond to timepoints in A), in six mice. (**H**) Change in ratio $V/D^2$, plotted across timepoints from F (V = velocity, D = diameter), in six mice (mean ±1 SD). Flow-rate is proportional to the product $V*D^2$. Since V and D were independently measured, and since change in flow in arterioles and venules was found to be conserved in G, plotting the ratio of the two shows the relative contributions of each independent variable to the change in flow, in arterioles and venules. This in effect shows the functionally opposing behaviours of the two vessel types in response to inflammation.

The online version of this article includes the following figure supplement(s) for figure 2:

**Figure supplement 1.** Variation between nasal and temporal retinal veins of the same eye during EIU, in six mice.

technique to the clinic. While some studies *Rossi et al., 2017* have used up to ~280 µW of NIR light, other studies *Liu et al., 2017* have used 115–130 µW of NIR light.

We also recognize the following challenges to translating our approach to use in human beings. The larger eye motion of the human eye without anesthesia will include saccades and drift that may impact image registration critical for stable time-lapse videography. Nevertheless, only a single image is required over any sparse time interval and therefore may be more lenient to this challenge. The investigation is rate-limited by the biology of immune cell dynamics, as most cells move relatively slowly, so will require patient cooperation over extended periods of imaging time. As we demonstrate, however, that single-cell surveillance can be observed over minutes, the approach would be feasible in a single imaging session, and thus still shorter than comparable tests in the clinic such as electroretinography. Next, on the imaging side, there will be ~4 times reduced axial imaging resolution in the human eye compared to the mouse eye due to the ~2 times smaller numerical aperture (*Geng et al., 2012*). This may impact the axial specificity or contrast of the single immune cell dynamics visualized in humans. However, other translucent cell types have been successfully imaged using human AOSLO (*Chui et al., 2012*; *Scoles et al., 2014*; *Sulai et al., 2014*; *Rossi et al., 2017*; *Scoles et al., 2017*), holding promise that our approach presented here can be translated to humans.

# Materials and methods

## Key resources table

| Reagent type (species) or resource | Designation | Source or reference | Identifiers | Additional information |
|---|---|---|---|---|
| Strain, strain background (*M. musculus*, males) | Wildtype C57BL/6J mice | The Jackson Laboratory | RRID:IMSR_JAX:000664, JAX Stock number 000664 | six to 12 weeks old |
| Strain, strain background (*M. musculus*, males) | Cx3cr1-GFP Cx3cr1^tm1Litt hemizygous mice | The Jackson Laboratory | RRID:IMSR_JAX:005582, JAX Stock number 005582 | six to 8 months old |
| Strain, strain background (*M. musculus*, males) | Tg(CD68-EGFP)1Drg hemizygous mice | The Jackson Laboratory | RRID:IMSR_JAX:026827, JAX Stock number 026827 | six to 12 weeks old |
| Antibody | Alexa Fluor 647 anti-mouse Ly-6G antibody (1A8 clone) for labeling (Mouse monoclonal) | BioLegend | Catalogue number: 127610 | 2 µg total prepared in 100 µl PBS |

## Mouse strains

All mice were sourced from The Jackson Laboratory (Bar Harbor, Maine, USA) and maintained at the University of Rochester in compliance with all guidelines from the University Committee on Animal Resources and according to the Association for Research in Vision and Ophthalmology statement for the Use of Animals in Ophthalmic and Vision Research. Mice were fed with standard laboratory chow ad libitum and housed under a 12 hr light-dark cycle. The following numbers, types, sex and

ages of mice were used: Six C57BL/6J mice (JAX stock number 000664, males, 6–12 weeks old) were analyzed for longitudinal study of immune cell motility and blood flow after induction of EIU with LPS (*Figure 2*). A seventh mouse was rejected from analysis due to anterior media opacities arising from EIU, at peak inflammation, as determined by expert users. Three C57BL/6J mice (males, 6–12 weeks old) were used for Ly6G antibody labeling (*Figure 1E*). Three *hCD68-GFP* mice (stock number 026827, males, 6–12 weeks old) were used for positive confirmation of fluorescent infiltrating monocytes and macrophages (*Figure 1F and H*). Four *Cx3cr1*-GFP hemizygotes (stock number 005582, males, 6–8 months old) were used for positive confirmation of fluorescent tissue-resident myeloid cells (*Figure 1G and I*). Further details are included in the transparent reporting form published.

## Mouse preparation for imaging

Mice underwent anesthetic induction with intraperitoneal Ketamine (100 mg/kg) and Xylazine (10 mg/kg) before maintenance on 1% v/v isoflurane and supplemental oxygen through a nose cone. Pupils were dilated with a single drop of 1% tropicamide and 2.5% phenylephrine (Akorn, Lake Forest, IL). Internal temperature was controlled using an external heating pad adjusted to maintain continuous 37°C with monitoring via a rectal probe electrical thermometer (Physiosuite, Kent). A custom rigid contact lens of 1.6 mm base curve, 3 mm diameter and +10 Dioptre correction (Advanced Vision Technologies, Lakewood, Colorado) was placed centrally on the cornea and lubrication of the eye maintained by aqueous lubricant (GenTeal, Alcon Laboratories, Fort Worth, TX) during imaging. The eye was imaged in free space meaning there was no physical contact with the AOSLO, ensuring no compression causing alteration of intraocular pressure.

## Endotoxin-induced uveitis model

Following anesthesia, intravitreal injection with a 34-gauge Hamilton microsyringe through the pars plana was used to deliver 0.5 ng of lipopolysaccharide (LPS) from *E. coli* 055:B5 (Sigma) in a 1 µL volume of phosphate buffered saline (PBS) (*Chu et al., 2016*). Only one eye of each mouse was injected and used for the study.

## Intravenous antibody labeling

2 µg of primary conjugated anti-mouse Ly6G-Alexa Fluor 647 (clone 1A8, Biolegend) diluted into 200 µl PBS were injected intravenously via tail vein 10 mins prior to imaging as previously published (*Marki et al., 2018*; *Woodfin et al., 2011*).

## AOSLO imaging

Mice were imaged with a custom adaptive optics scanning light ophthalmoscope (AOSLO), using near-infrared light (796Δ17 nm, 200–500 µW, super luminescent diode: S790-G-I-15, Superlum, Ireland) (*Joseph et al., 2019*; *Geng et al., 2012*). The measured imaging spatial resolution was: lateral: 0.77 µm for 520 nm fluorescence and 1.2 µm for 796 nm reflectance (*Joseph et al., 2019*; *Geng et al., 2012*). Similarly, the confocal axial resolution was 10.5 µm for 520 nm fluorescence. For phase-contrast imaging, the axial depth of focus is narrow in the mouse (numerical aperture = 0.49) and provides optimal contrast in a thin optical section less than the diameter of most retinal cell types as described by us previously (*Guevara-Torres et al., 2020*). Given this numerical aperture, nearly 50 diopters of focus covers the mouse retina meaning that cells outside the plane of optimal focus will be rendered invisible due to defocus blur. Phase-contrast imaging referred to in the context of this paper, was achieved by purposefully displacing the detector axially to a plane conjugate to the highly reflective RPE/choroid complex, to enable detection of forward and multiply scattered light from translucent cells, as detailed in our recent publication (*Guevara-Torres et al., 2020*). In a subset of experiments for confirmation of immune cell types, fluorescence was simultaneously imaged using 488 nm excitation (220-330 µW) and 520Δ35 emission for GFP, and 640 nm excitation and 676Δ29 emission for Alexa Fluor 647 (excitation laser diode: iChrome MLE, Toptica Photonics, Farmington, New York, USA; emission filters: FF01-520/35-25 and FF01-676/29-25, Semrock, Rochester, New York, USA). For AOSLO data acquisition and electronics control, we implemented a Xilinx ML605 Virtex-6 based FPGA circuit board (Xilinx Inc, San Jose, CA) integrated with two Analog Device EVAL-AD9984AEBZ circuit boards (Analog Device Inc, Cambridge, MA) for up to six channels

of data acquisition from the AOSLO; two imaging channels were used for this study. Mice also underwent imaging with HRA+OCT Spectralis (Heidelberg Engineering, Germany). Reversible severe vitritis or cataract precluding AOSLO imaging occurred at a peak inflammation timepoint in 25% of eyes induced with EIU in our work.

For simultaneous fluorescence and NIR phase-contrast imaging we corrected for LCA using independent focusing of the different imaging channels (*Geng et al., 2012*). A calibration mouse was used with fluorescein labelled blood plasma to align the visible and NIR channels to the same anatomical locations/focal planes (*Joseph et al., 2019*). TCA too was corrected online using small shifts of the entrance pupil, and offline during registration (*Granger et al., 2018*).

### Aberration measurement and correction with adaptive optics

Aberrations in the mouse eye were measured and corrected with a closed-loop adaptive optics system operating at 13 corrections per second, built at the University of Rochester and described previously (*Joseph et al., 2019*; *Geng et al., 2012*). Aberrations were measured with a Hartmann-Shack wavefront sensor using a 904 nm wavefront beacon (QFLD-905–10S, QPhotonics, Ann Arbor, Michigan, USA) imaged onto the retina. Aberration correction was achieved with a continuous membrane deformable mirror with 97 actuators (DM-97–15, ALPAO, France).

### Imaging videography

The AOSLO is a raster-scanning instrument with a resonant scanner frequency of 15 kHz and 25 Hz orthogonal scanning rendering the retina at 25 frames per second. Point scanning readout was achieved by two photomultiplier tubes (PMTs) for visible and near infrared wavelengths (H7422–40 and H7422–50, Hamamatsu, Japan). Frame size was 608 by 480 pixels and the image distortion introduced by sinusoidal scanning was corrected in real-time (*Yang et al., 2015*). Field sizes were between 2 and 5 degrees of visual angle corresponding to 68–170 microns in retinal space. Typical imaging sessions lasted ~2 hr and semi-continuous video acquisition of a target retinal location was conducted for up to 30 min. AOSLO retinal data was visualized in real-time to facilitate user tracking, correction and optimization and saved for subsequent post-processing.

### Image registration and time-lapse analysis

To correct for residual motion of the eye, image registration was performed with custom software (*Yang et al., 2015*; *Dubra and Harvey, 2010*). Time-lapse videos were generated using running frame-averaging of 25–50 frames (from 25 Hz native frame rate of AOSLO) with ImageJ and Fiji (National Institutes of Health, USA) (*Schindelin et al., 2012*). Montaging multiple fields in *Figure 1a* was performed by stitching and blending overlapping 4.98 degree fields manually using Adobe Photoshop (Version: CS6 Extended v13.0.1 $\times$ 64).

### Statistical analysis

Pearson correlation and Freidman tests were performed using Prism 7.0 (GraphPad Software) and linear curve-fitting in MATLAB 2020a, version 9.8.0.

### Cell migration measurement

Retinal locations were imaged for 100 s for six mice at five timepoints (baseline, 6, 24, 72 hr and 10 days post LPS injection). Sixty AOSLO phase-contrast videos were used to analyse the migration of extravasated cells. Registered videos were pre-processed by cropping to 512 $\times$ 400 pixels and temporally averaged with five frames. A customized semi-automated deep-learning based cell tracking software was employed to track and quantify the migration behavior of cells. The software consisted of a deep learning-based cell detector with an encoder-decoder U-Net backbone architecture (*Falk et al., 2019*; *Tsai et al., 2019*). The U-Net was trained on phase-contrast AOSLO images with the centroids of 387 cells manually identified by expert graders obtained from four mice at either 6 or 24 hr post LPS injection. Immune cell data from mice imaged for training were excluded from the final analysis of the results in this paper. The trained U-Net outputs a probability map for each frame of a video that was thresholded (>=90%) to identify the centroids of cells. The cell counts of a video were calculated by averaging the number of detected objects in the first 25 frames (5 s). To track the cells, centroid positions detected by the U-Net in adjacent frames were linked with a nearest

neighbor search algorithm (*Tsai et al., 2019*). The deep learning strategy facilitated tracking of a large number of cells across multiple frames captured at different time points across inflammation. Once the heavy burden of tracking this population over a large data set was complete, a human user provided quality control of the automated outputted traces to confirm tracking fidelity. Traces due to incorrect linkage (for example, a single trace that falsely jumped between two adjacent cells) were manually rejected based on visual inspection (*Figure 2b*, *Video 8*). To quantify cell migration, two quantities were extracted from the traces: 1. cell displacement which is defined as the displacement of a cell over 100 s. 2. confinement ratio which is defined as the ratio of cell displacement and the total path length over 100 s. Traces in *Figure 2c* indicate the total positional movement and direction of movement relative to cell position in the first second of data collection. The U-Net based cell detection was performed with Python 3.7 and PyTorch 1.0.1, while the tracking and quantification procedures were based on MATLAB.

## Blood flow measurement

Single cell blood flow was imaged and measured with near infrared light using our recently published approach (*Joseph et al., 2019*), for the same six mice and five timepoints as above, in an arteriole and venule surrounding the tissue location at which the cell migration measurements above were done. Briefly, a fast 15 kHz beam (796Δ17 nm) was scanned across a vessel of interest to image passing blood cells without requiring contrast agent. Cellular-scale blood velocity and vessel diameter were quantified automatically. Given the small size of even the largest mouse retinal vessels (<45 µm inner diameter in healthy mice), the spatial resolution of our approach accurately measured the inner lumen diameter with micrometer precision, accounting for vessel tortuosity, vascular wall thickness and cell-free plasma layer. Additionally, the temporal resolution of the velocity detection approach was more than sufficient to measure and account for cardiac pulsatility in flow, as demonstrated previously. This ensured accurate measurement of the average blood velocity through the vessel. Combined, the volumetric flow rate through the vessel was quantified label-free. The non-invasive approach enabled us to track blood flow longitudinally from hours to weeks over the course of inflammation without requiring invasive injections or euthanasia after a single timepoint was imaged.

## Code and data availability statement

Single-cell blood flow was measured using our recently published approach (*Joseph et al., 2019*), with custom code written in MATLAB R2017a (Version 9.2, with Image Processing Toolbox, MathWorks, MA). Source code is available in public repository here: *Joseph, 2020* https://github.com/abyjoseph1991/single_cell_blood_flow (copy archived at https://github.com/elifesciences-publications/single_cell_blood_flow_2). AOSLO dataset has been made available at a public repository here: https://doi.org/10.5281/zenodo.2658767.

## Acknowledgements

The authors thank Qiang Yang, Andres Guevara, Rachel Hollar, Jennifer Strazzeri and Karteek Kunala for their technical contributions to this work. We are grateful to Andrew Dick and Richard Lee for their guidance and helping establish this collaboration. We thank David Williams, Ashwath Jayagopal and Robin Sharma for critical feedback on the manuscript and project suggestions.

## Additional information

### Competing interests

Aby Joseph: Received funding support from Hoffman-La Roche Inc Hold patents and/or patent applications on adaptive optics technology filed through the University of Rochester. US Patent Application Number: US20190114790A1, "Rapid assessment and visual reporting of local particle velocity". Guanping Feng, Kosha Dholakia: Received funding support from Hoffman-La Roche Inc. Jesse Schallek: Received funding support from Hoffman-La Roche Inc Hold patents and/or patent applications on adaptive optics technology filed through the University of Rochester. US Patent

#9,844,320 Issued: 12/19/2017, "System and Method for Observing an Object in a Blood Vessel". US Patent Application Number: US20190114790A1, "Rapid assessment and visual reporting of local particle velocity". The other author declares that no competing interests exist.

## Funding

| Funder | Grant reference number | Author |
|---|---|---|
| National Institutes of Health | R01 EY028293 | Aby Joseph<br>Guanping Feng<br>Kosha Dholakia<br>Jesse Schallek |
| National Institutes of Health | P30 EY001319 | Aby Joseph<br>Guanping Feng<br>Kosha Dholakia<br>Jesse Schallek |
| Research to Prevent Blindness | Unrestricted grant to Department of Ophthalmology | Aby Joseph<br>Guanping Feng<br>Kosha Dholakia<br>Jesse Schallek |
| Research to Prevent Blindness | a Career Development Award and Stein Award | Aby Joseph |
| Roche | Research grant | Aby Joseph<br>Guanping Feng<br>Kosha Dholakia<br>Jesse Schallek |
| Dana Foundation | David Mahoney Neuroimaging Award | Jesse Schallek |
| NIHR | Academic Clinical Lecturer | Colin J Chu |
| National Eye Research Centre | | Colin J Chu |
| Worldwide Universities Network | Research Mobility Programme Award | Colin J Chu |

The funders had no role in study design, data collection and interpretation, or the decision to submit the work for publication.

## Author contributions

Aby Joseph, Conceptualization, Data curation, Software, Formal analysis, Investigation, Visualization, Methodology, Writing - original draft, Writing - review and editing; Colin J Chu, Conceptualization, Resources, Data curation, Funding acquisition, Investigation, Visualization, Methodology, Writing - original draft, Writing - review and editing; Guanping Feng, Software, Formal analysis, Visualization, Writing - original draft; Kosha Dholakia, Data curation, Investigation; Jesse Schallek, Conceptualization, Resources, Supervision, Funding acquisition, Methodology, Writing - review and editing

## Author ORCIDs

Aby Joseph https://orcid.org/0000-0001-8143-801X
Colin J Chu https://orcid.org/0000-0003-2088-8310
Jesse Schallek https://orcid.org/0000-0002-6337-4187

## Ethics

Animal experimentation: All guidelines of University Committee on Animal Resources at the University of Rochester, Rochester, New York, USA, were followed. PHS Assurance #D16-00188(A3292-01). Reference number from University Committee on Animal Resources is 101017 and protocol number is 2010-052. Mice were treated according to the Association for Research in Vision and Ophthalmology Statement for the Use of Animals in Ophthalmic and Vision Research.

## Decision letter and Author response

Decision letter https://doi.org/10.7554/eLife.60547.sa1

Author response https://doi.org/10.7554/eLife.60547.sa2

---

## Additional files

### Supplementary files
• Transparent reporting form

### Data availability

Dataset has been made available at a public repository (Zenodo) here: https://doi.org/10.5281/zenodo.2658767.

The following previously published dataset was used:

| Author(s) | Year | Dataset title | Dataset URL | Database and Identifier |
|-----------|------|---------------|-------------|-------------------------|
| Joseph A, Guevara-Torres A, Schallek J | 2019 | AOSLO Single Cell Blood Flow - Raw Data (eLife paper: Joseph et al. 2019) | https://doi.org/10.5281/zenodo.2658767 | Zenodo, 10.5281/zenodo.2658767 |

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
