## [Decision Letter]

**Acceptance summary:**

The authors present a novel adaptive optics-based method to visualize the motion of immune cells in the mouse retinal circulation without cell labeling. This simultaneous non-invasive imaging of blood flow and immune cells opens up new possibilities for investigating the role of the vasculature in inflammation and immune responses.

**Decision letter after peer review:**

Thank you for submitting your article "Label-free imaging of immune cell dynamics in the living retina using adaptive optics" for consideration by *eLife*. Your article has been reviewed by three peer reviewers, including Johnny Tam as the Reviewing Editor and Reviewer #1, and the evaluation has been overseen by Anna Akhmanova as the Senior Editor. The following individuals involved in review of your submission have agreed to reveal their identity: Timothy Secomb (Reviewer #2); Tyson Kim (Reviewer #3).

The reviewers have discussed the reviews with one another and the Reviewing Editor has drafted this decision to help you prepare a revised submission.

The reviewers raised some specific and useful suggestions. I have combined them into a single review below to help improve the clarity and presentation of the work.

Summary:

Building upon their prior *eLife* article, the authors have extended their approach to image immune cells that are most easily seen in an endotoxin-induced mouse model of uveitis. The authors present a novel approach to directly image the motion of immune cells in the mouse retinal circulation, without the use of dyes or contrast agents. The ability to carry out longitudinal non-invasive imaging alongside confirmatory experiments with fluorescently-tagged cells is compelling. This simultaneous non-invasive imaging of blood flow and immune cells in the mouse retinal circulation opens up new possibilities for investigating the role of the vasculature in inflammation and immune responses. The work is clearly and concisely presented.

Essential revisions:

1) The novelty and impact of this manuscript is based on the specific implementation and potential for human application. The authors allude to human imaging in the Introduction and Conclusion, and often describe their work in a broader context of ocular imaging. However, there is currently inadequate description of relevance and consideration for human use. The authors should include a robust description of potential human applications and limitations in the Discussion section of the manuscript. This imaging method may provide an exquisite level of quantitative assessment in retinitis, uveitis, and ocular lymphoma. It may also offer immune surveillance in a number of systemic diseases.

2) In order to better understand the variation in cells observed both across difference mice as well as in different areas within an eye, it is suggested to clarify whether any variability was observed across the different mouse eyes, and if available, across the different areas imaged in the eye. For example, in Figure 1, it's not clear if this impressive degree of cells is observed on the other veins too. Clarifying the variability would be extremely helpful for documenting how representative this data is for the purposes of maximizing repeatability. In the *eLife* transparent reporting form, specifying the number/type of each mice used for each experiment in addition to the measurement replication and technical replicates would be useful.

3) It's possible that not all cells are fluorescently tagged (especially in the case of the Ly6G-Alexa Fluor 647 tail vein injection). Presumably, only those cells that are in the proximity of the injection bolus will be efficiently labeled. This should be clarified in the manuscript.

4) Can the authors say anything about why microscopy at this particular wavelength shows up the immune cells?

5) The authors should comment on the ability/inability to image deeper structure including the subretinal space and choroid, which are highly involved in many immune-mediated pathologies of the eye.

6) How might this imaging method perform with vitritis and media opacity, which are often important features of uveitis? It would be valuable to describe or show data on the performance of this imaging approach with media opacity (vitritis, cataract).

7) The authors perform imaging with near infrared light (796nm) at power levels of 200-500 microwatts. Given the implications for human application and the broad readership of *eLife*, the authors should comment on prospective safety in human application, such as for the power levels of near infrared light (796 nm, 200-500 microwatts).

---

## [Author Response]

Essential revisions:1) The novelty and impact of this manuscript is based on the specific implementation and potential for human application. The authors allude to human imaging in the Introduction and Conclusion, and often describe their work in a broader context of ocular imaging. However, there is currently inadequate description of relevance and consideration for human use. The authors should include a robust description of potential human applications and limitations in the Discussion section of the manuscript. This imaging method may provide an exquisite level of quantitative assessment in retinitis, uveitis, and ocular lymphoma. It may also offer immune surveillance in a number of systemic diseases.

We agree with the idea that there is tremendous clinical potential and we are grateful for the encouragement to highlight this exciting future prospect. We have expanded the Discussion for a more robust description of human applications and limitations.

*“*There are numerous benefits if this approach can be applied to the human eye. […] However, other translucent cell types have been successfully imaged using human AOSLO (Rossi et al., 2017, Chui et al., 2012, Scoles et al., 2014, Scoles et al., 2017), holding promise that our approach presented here can be translated to man.”

As addressed in revisions above, we agree that there is great interest in human application. However, we would also highlight that the novelty and impact of this work is applicable to visualizing CNS immune cells in one of the most widely used models in biomedical science to date, the mouse. We feel the readership will also appreciate that this approach facilitates a far wider range of science applications than only the clinically translational potential.

2) In order to better understand the variation in cells observed both across difference mice as well as in different areas within an eye, it is suggested to clarify whether any variability was observed across the different mouse eyes, and if available, across the different areas imaged in the eye. For example, in Figure 1, it's not clear if this impressive degree of cells is observed on the other veins too. Clarifying the variability would be extremely helpful for documenting how representative this data is for the purposes of maximizing repeatability. In the eLife transparent reporting form, specifying the number/type of each mice used for each experiment in addition to the measurement replication and technical replicates would be useful.

For demonstration and brevity, we had originally placed only a few visual examples of venous infiltrating cells (Figure 1). However, we agree that the argument is more compelling with multiple examples. Therefore, we provide a new supplementary figure (Figure 2—figure supplement 1): a catalog of infiltrating cells of six mice in veins on both sides of the optic disc.

Furthermore, like most retinal insults or chronic disease, the manifestation of the disease phenotype is geographic. Variation exists between veins of the same eye and within eyes from different mice (inter and intraocular variation). The variation could be accounted for by known variability of the EIU model and biological response (Chu et al., 2016). While this variation may initially appear to reduce “repeatability”, we contend that the power of our approach is this very ability to focus in on a single retinal location and track it longitudinally without surgery or histology, to provide an internal control. This is the basis of our data in Figures 1 and 2 of the paper and is perhaps the greatest testament to reducing experimental variability within cohorts, post-mortem artifact, and the aforementioned phenotype variability that is present in most retinal disease.

To the reviewers’ further comments, the Materials and methods section has been updated:

“The following numbers, types, sex and ages of mice were used: Six C57BL/6J mice (JAX stock number 000664, males, 6-12 weeks old) were analyzed for longitudinal study of immune cell motility and blood flow after induction of EIU with LPS (Figure 2). […] Four Cx3cr1-GFP hemizygotes (stock number 005582, males, 6-8 months old) were used for positive confirmation of fluorescent tissue-resident myeloid cells (Figure 1G and I).”

Numbers and types of mice have also been added to Figure 1 legend as:

“Representative images from cohort of eleven (A to D, J) and three C57BL/6J mice (E), three CD68-GFP mice (F and H) and four Cx3cr1-GFP mice (G and I).”

These numbers have also been summarized in the *eLife* transparent reporting form as requested by the reviewers.

3) It's possible that not all cells are fluorescently tagged (especially in the case of the Ly6G-Alexa Fluor 647 tail vein injection). Presumably, only those cells that are in the proximity of the injection bolus will be efficiently labeled. This should be clarified in the manuscript.

We concur with this limitation and the manuscript text has been amended (see below). The antibody is known to be specific, so fluorescent cells are almost certain to be neutrophils, yet we rightly cannot be certain of the identity of the unlabeled population in this context and have altered the text. For reader’s reference, previous work has shown the main population in EIU are neutrophils (Chu et al., 2016) and the technique is widely used, yet one of the few studies to show data on labelling efficiency identifies only 10% of circulating leukocytes are tagged. (Bucher et al., 2015).

Text altered to:

“Simultaneous phase contrast and confocal fluorescence AOSLO revealed most leukocytes rolling along venular endothelium were neutrophils using intravenous anti-Ly6G fluorescent antibody labelling, though the proportion may be higher as only 10% of circulating leukocytes may be labelled using this method (Bucher et al., 2015) (Figure 1E, Video 5 ).”

4) Can the authors say anything about why microscopy at this particular wavelength shows up the immune cells?

Our choice of imaging wavelength was driven by the following considerations: First, NIR (near infrared) light (796 nm) provided lower phototoxicity regimes for imaging the retina than visible light. This means that higher light levels of NIR can be used without phototoxicity or subject discomfort. This is a similar strategy employed by many ophthalmoscopy approaches such as NIR fluorescence imaging, commercial OCT and SLO. Second, the use of NIR also provides optical benefit of greater theoretical penetration into the retinal tissue without scatter typical of short visible wavelengths and was favorably transmitted by the anterior optics and cells of the retina with minimal absorption.

We also point the readers to our recent publication (Guevara-Torres et al., 2020) that provides a detailed wave-optics simulation to determine the source of contrast in this modality using this particular wavelength (796 nm). In those simulations, we found that the combination of this wavelength and other imaging parameters rendered contrast to a large catalog of translucent retinal cells.

Having stated this, we have not tested other wavelengths for phase-contrast imaging, and expect there to be a tradeoff of light-budget, scatter and subject comfort using shorter wavelength (visible) light. Despite a few theoretical benefits, we are reluctant to state more in the manuscript as it would be speculative and not based in scientific rigor.

In summary, we agree with the need to highlight these aspects. We summarize the above and address the reviewers’ comments in the revised manuscript:

“Our choice of imaging wavelength (796 nm) provided both minimal phototoxicity and greater theoretical penetration through the eye’s anterior optics and minimal absorption in the retina without scatter typical of short visible wavelengths. Furthermore, we have previously published a detailed wave-optics simulation to determine the source of contrast in our imaging modality (Guevara-Torres et al., 2020), where we found that the combination of this wavelength and other imaging parameters rendered reasonable contrast to translucent cells.”

5) The authors should comment on the ability/inability to image deeper structure including the subretinal space and choroid, which are highly involved in many immune-mediated pathologies of the eye.

Agreed. Our phase-contrast approach relies on imaging cells in front of the backscattered light from RPE/choroid complex (described in detail by Guevara-Torres et al., 2020). As such, we expect contrast will be lower when attempting to image the subretinal and choroidal space due to scatter/absorption of the RPE. Importantly though, these values are expected to be non-zero (as measured and modeled by Guevara-Torres et al., 2020).

In the current manuscript, we focus upon inflammation of the inner retina, vasculature and peripapillary region, which is still frequently involved in many forms of ocular inflammation. While it might seem challenging to image deeper cell structures, we draw the readers’ attention to Scoles et al., 2017, (we thank the reviewers for highlighting this omission), which uses a similar approach to visualize a dynamic structure suspected to be a macrophage next to the photoreceptor layer of the human retina, in response to retinal disease. Whilst the cell type could not be confirmed, it raises the possibility that in certain contexts deeper retinal imaging of immune cells with phase contrast approaches could be possible, even in humans.

To address the above, we have revised the Discussion and included the Scoles et al., 2017 reference:

“A further limitation is that deeper retinal structures are not as well visualized due to the nature of contrast (Guevara-Torres et al., 2020). […] Our work reinforces this exciting finding as we now demonstrate, with transgenic markers in the mouse, that tissue immune cells are reliably detectable by label-free phase-contrast AOSLO.”

6) How might this imaging method perform with vitritis and media opacity, which are often important features of uveitis? It would be valuable to describe or show data on the performance of this imaging approach with media opacity (vitritis, cataract).

We agree with the reviewers that this is an important point and address it in the following ways. First, as mentioned in point 4 above, this is an additional benefit of using NIR light as anterior scatter may be mitigated to a small extent. However, severe vitritis, cataract or other significant media opacities do affect almost all ophthalmological imaging approaches that use light and therefore will be a formidable hurdle to overcome. AO imaging is subject to the same degradation due to scatter.

Our new supplementary figure, Figure 2—figure supplement 1, provides examples of the variation of retinal inflammation in the mouse where imaging is still possible. We expect the initial deployment of this technique will find greatest success in the study of primary retinal inflammation. Many types of posterior uveitis do not develop severe vitritis or anterior chamber inflammation that would result in prohibitive loss of optical clarity. In cases that do, imaging soon after the initiation of treatment when media opacity improves can still provide important biological insight.

The occurrence of vitritis limiting imaging in our cohorts has now been included in the transparent reporting form. We have also added the following clarifications to the manuscript.

Discussion:

“Consistent with all AOSLO approaches, in our approach, relatively clear optical media is required and severe cataract or vitritis during peak inflammation can limit successful imaging. In our study, this affected 25 % of the eyes given EIU.”

Materials and methods:

“Reversible severe vitritis or cataract precluding AOSLO imaging occurred at a peak inflammation timepoint in 25% of eyes induced with EIU in our work.”

Of note, however, the EIU model arguably develops more severe vitritis than seen in most diseases or disease models and induces inflammation (including hypopyon) of the anterior optics (Chu et al., 2016). Our images show empirically that it is still possible to get sufficient image contrast and resolution in many cases.

7) The authors perform imaging with near infrared light (796nm) at power levels of 200-500 microwatts. Given the implications for human application and the broad readership of eLife, the authors should comment on prospective safety in human application, such as for the power levels of near infrared light (796 nm, 200-500 microwatts).

Thank you for highlighting this. We have included the following new text in revised manuscript:

“Our approach is in a similar power range to those already employed safely in human AOSLO studies, where phase contrast approaches have also been used to study neuronal and photoreceptor cells using a similar light source and wavelength (Rossi et al., 2017, Scoles et al. 2014). […] While some studies (Rossi et al. 2017) used up to ~280 µW of NIR light, other studies (Liu et al. 2017) have used 115-130 µW of NIR light.”

In addition to the above edits in the manuscript, we draw the reviewers’ attention to Rossi et al., 2017 for precedence in compliance with the ANSI standard in the human retina. While we have provided this for clarity for the readership, we are reluctant to state more in the manuscript as the ANSI light safety standard applies only to the human eye and it is difficult to apply it directly to the mouse eye due to species difference, photopigment density, subtended visual angle, visual acuity limits, lack of fovea and strong pigmentation of the RPE, among others, that make a direct comparison between human and mouse challenging.